# Heads up–Four *Giraffa* species have distinct cranial morphology

**Nikolaos Kargopoulos**[1,2]*, **Jesús Marugán-Lobón**[1,3,4], **Anusuya Chinsamy**[1], **Bernard R. Agwanda**[5], **Michael Butler Brown**[2], **Stephanie Fennessy**[2], **Sara Ferguson**[2], **Rigardt Hoffman**[2], **Fredrick Lala**[6], **Arthur Muneza**[2], **Ogeto Mwebi**[5], **Moses Otiende**[6], **Alice Petzold**[7,8], **Sven Winter**[9,10], **Abdoul Razack Moussa Zabeirou**[2,11], **Julian Fennessy**[2,12]

**1** Department of Biological Science, University of Cape Town, Cape Town, South Africa, **2** Giraffe Conservation Foundation, Windhoek, Namibia, **3** Department of Biology, Unidad de Paleontología, Universidad Autónoma de Madrid, Madrid, Spain, **4** CIPb-UAM, Center for the Integration in Paleobiology, Universidad Autónoma de Madrid, Madrid, Spain, **5** National Museums of Kenya, Nairobi, Kenya, **6** Wildlife Research and Training Institute, Naivasha, Kenya, **7** Institute of Biochemistry and Biology, University of Potsdam, Potsdam, Germany, **8** Museum für Naturkunde—Leibnitz Institute for Evolution and Biodiversity Science, Berlin, Germany, **9** Senckenberg Biodiversity and Climate Research Centre, Frankfurt am Main, Germany, **10** Research Institute of Wildlife Ecology, University of Veterinary Medicine, Vienna, Austria, **11** Faculty of Tropical AgriSciences, Czech University of Life Sciences Prague, Prague, Czech Republic, **12** School of Biology and Environmental Science, University College Dublin, Dublin, Ireland

\* nikoskargopoulos@gmail.com

**Data Availability Statement:** Data are available on https://github.com/nkargopoulos/Giraffe-Cranial-3D-GM.git.

## Abstract

Giraffe (*Giraffa* spp.) are among the most unique extant mammals in terms of anatomy, phylogeny, and ecology. However, aspects of their evolution, ontogeny, and taxonomy are unresolved, retaining lingering questions that are pivotal for their conservation. We assembled the largest known dataset of *Giraffa* skulls (n = 515) to investigate patterns of cranial variability using 3D geometric morphometrics. The results show distinct sexual dimorphism and divergent ontogenetic trajectories of skull shape for the north clade (*G. camelopardalis antiquorum*, *G. c. camelopardalis*, *G. c. peralta*, and *G. reticulata*) and the south clade (*G. giraffa angolensis*, *G. g. giraffa*, *G. tippelskirchi tippelskirchi*, and *G. t. thornicrofti*) which was further supported statistically. Discriminant functions found statistically significant cranial shape differences between all four *Giraffa* species, and in some cases also between subspecies of the same species. Our 3D morphometric analysis shows that the four genetically distinct *Giraffa* spp. also have distinct cranial morphologies, largely addressable to features of display (ossicones). Our results highlight the importance of focusing future giraffe conservation efforts on each taxon to maintain their unique characteristics and biodiversity in the wild.

## Introduction

Taxonomy is one of the most significant disciplines of biology in setting the framework for many other scientific branches, including conservation [1–5]. However, the concept of species, the most fundamental unit in taxonomy, is far from well-defined. New approaches, access to

**Funding:** The author(s) received no specific funding for this work.

**Competing interests:** The authors have declared that no competing interests exist.

technology, and increased data have contributed to more complex views on the distinction between taxa [6]. The present study on the cranial structure of extant *Giraffa* is of considerable scientific interest due to the uniqueness of the giraffid anatomy, as well as the relevance of the results to the conservation of giraffe. Firstly, giraffid ossicones are unique anatomical structures not observed in any other extant genera except for the four *Giraffa* species and their closest relative, the okapi (*Okapia johnstoni*), which is markedly different than the giraffe in terms of ossicone morphology [7–13]. Over the past two centuries, *Giraffa* taxonomy has been debated, in part due to its relative phylogenetic isolation [14–16], and many nomenclatural frameworks have been suggested with ossicones used as a prominent descriptive and diagnostic feature [17–37]. These taxonomic inconsistencies have had direct implications on conservation, given that species serve as the main conservation unit for organizations such as the International Union for the Conservation of Nature (IUCN) and governments [1–5, 29, 30]. As such, evidence supporting alternative taxonomic classifications may have significant implications for *Giraffa* conservation policies and management across their respective range states. Moreover, differences in cranial shape, especially regarding ossicone form disparity, is an important factor that affects giraffe reproductive behavior [28, 38, 39]. Consequently, ossicone morphology bears high importance in further understanding *Giraffa* taxonomy with direct implications for their ecology, behavior, evolution, and conservation, making the results of the present study a significant input for taxonomic discussions.

The morphology of *Giraffa* ossicones was used for taxonomic purposes in several previous studies [8, 18, 19, 23]. Seymour [18] evaluated cranial morphology of a smaller giraffe dataset using an array of different methodologies, including 2D Geometric Morphometrics. The results of the latter study similarly showed clear sexual dimorphism between the north and south clades based mainly on the median ossicone, while changes in the frontal ossicones and the rostrum were also observed. However, despite observing consistent discrimination in the pelage, skeleton, and mitochondrial genetics of the different taxa, Seymour [18] followed a lumping approach, akin to Singer and Boné [17] and Dagg [27, 40], in interpreting this wide morphological range with partial overlaps as one species with geographical variants treated as subspecies. In this study, we conducted a wider comparison using significantly more specimens of all *Giraffa* taxa, utilizing state-of-the-art software, and digitizing 3D coordinates, revealing statistically significant differences between the studied groups (Fig 1).

## Material and methods

### Material

The studied material consists of a dataset of 515 *Giraffa* skulls. To our knowledge, this is the largest dataset of extant *Giraffa* skulls ever assembled. A summary of the specimen numbers per taxon is presented in S1 Table. The skulls were sourced from museum collections (n = 363, 70.5%), wild samples in parks/reserves throughout Africa (n = 70, 13.6%), and taxidermy companies with samples from various known parks/reserves in southern Africa (n = 82, 15.9%). No permits were required for the described study, which complied with all relevant regulations. A detailed catalogue of the studied specimens is available in S2 Table and a list of institutional abbreviations in S1 Appendix. Since the geographical overlap between the different taxa is limited, provenance was used to a priori identify the taxonomy of each specimen.

### Methods

**Digitization and visualization.** Most specimens were 3D scanned using a EinScan Pro HD handheld surface scanner. The specimens from NER-KOU were CT scanned in Polyclinique Magori (Niamey, Niger; slice 0.75, kernel B20s). The specimens housed in NMB were

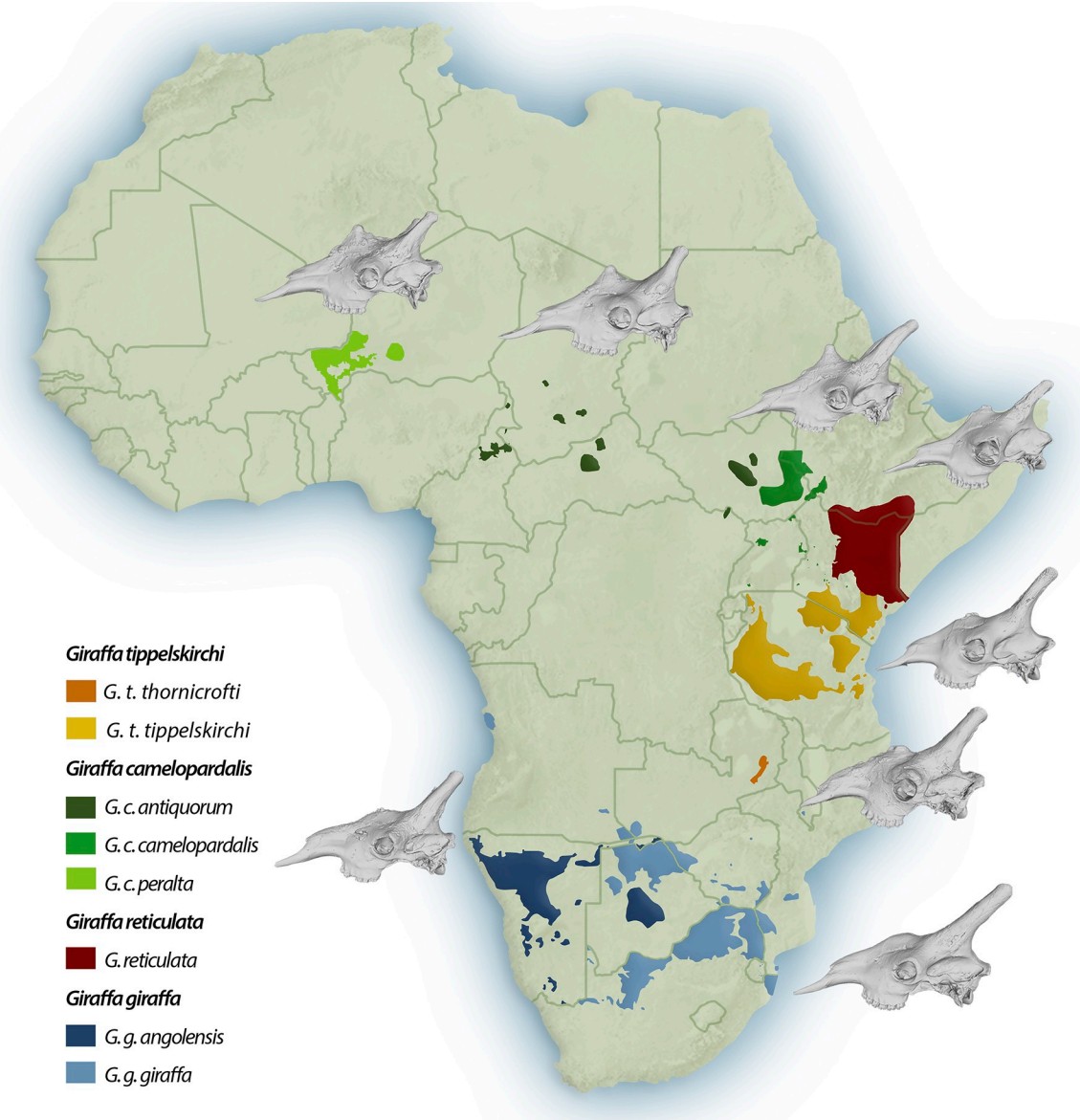

**Fig 1. Map showing the geographical range of the extant giraffe species and subspecies as well as representative male skulls of each subspecies in lateral view.**

scanned by Luca Pandolfi using an Artec Eva 3D scanner and were kindly provided to us for the present paper. The specimens in ZFMK were scanned using an Artec Space Spider 3D scanner, whereas the specimens in ZMB and SA-SI were scanned using an Artec Leo 3D scanner. CT-scanned specimens were segmented using 3D Slicer [41].

**3D geometric morphometrics.** Landmarking on the digital skulls was performed using Stratovan Checkpoint [42] following the protocol shown in S3 Table. The landmark protocol covers most of the salient features of the giraffe skull anatomy and includes 42 single fixed landmarks, as well as 8 curves that are defined by 31 fixed landmarks and 46 semilandmarks, with two semilandmarks between each pair of fixed landmarks in the curves (Fig 2).

Landmark coordinates were imported into the R package *geomorph* 4.0.7 [43] for the estimation of missing landmarks and sliding of semilandmarks and the subsequent Generalized

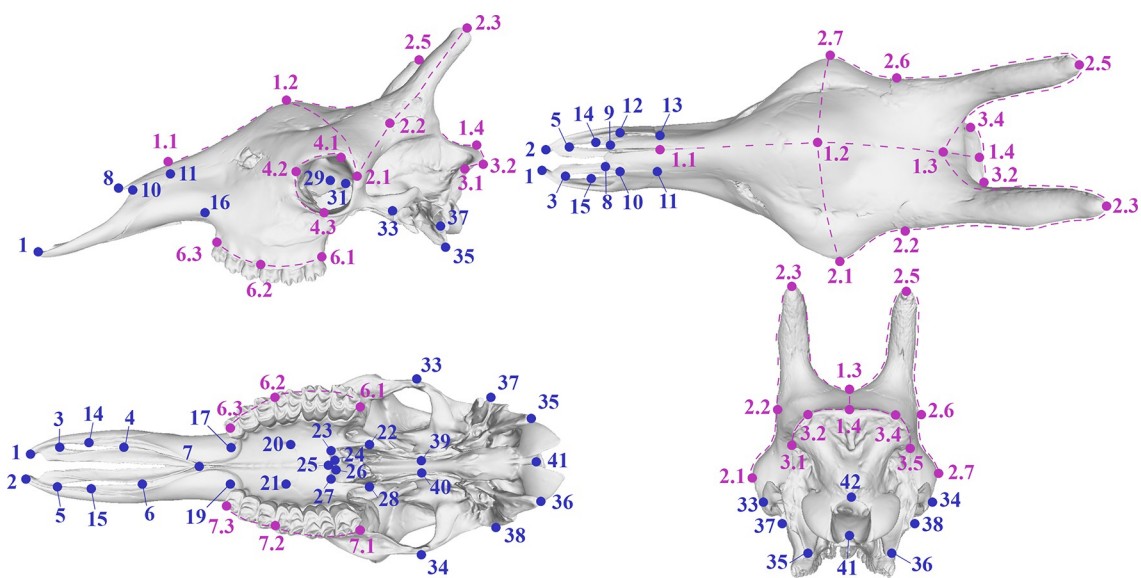

**Fig 2. The landmark protocol used in this study.** Single points are marked blue, whereas lines are marked purple. Detailed descriptions of the landmarks can be found in S3 Table.

Procrustes Analysis. The resulting Procrustes Coordinates (utilizing the Bending Energy Criterion) and Centroid Sizes were exported for further analysis. The code for GPA and the subsequent phylomorphospace can be found on https://github.com/nkargopoulos/Giraffe-Cranial-3D-GM.git.

The Procrustes Coordinates were imported into MorphoJ [44], which was used to conduct Multiple Regression between the Symmetric Component and Centroid Size (exported from R), Principal Component Analysis, Canonical Variate Analyses, and Discriminant Function Analyses to reveal the relationships between the different groups of the dataset. Scores of these results were imported in PAST 4.16 [45] to visualize the plots. The shape changes of the analyses were visualized in Landmark Editor [46] through 3D skull morphing targeted to the respective coordinates exported from MorphoJ using as a reference the specimen RMCA-RG-25548 (*Giraffa camelopardalis antiquorum*, female, age class I). Phylomorphospaces were created in R *phytools* 2.3–0 [47] based on phylogenetic trees created in Mesquite 3.81 [48] based on the phylogenetic hypothesis proposed by Coimbra et al. (2021) [34].

**Age classes.** A protocol was developed to group the specimens into age categories based on dental wear inspired by the study of Hall-Martin [49]. Since the ontogenetic changes of the giraffe skull are studied through Geometric Morphometrics, the a priori age classes did not include characteristics associated with skull shape, as that would lead to circular reasoning. As such, only changes of the surface of the skull were chosen, more specifically eruption of teeth, dental wear, secondary ossification, and suture closure. The age classes can be seen in S4 Table.

The absence of age data on specimens from the wild inevitably restricts the formation of any direct age protocol. Therefore, it is possible that specimens attributed to an age class might in fact be slightly younger or older than that. Note that dental wear can show different results in the two sides of the maxilla. For example, in the specimen NMW– 26429/ST 292 the right M1 does not have a dentine connection between the protocone and the metacone (age class F), whereas its left side did (age class G). In such cases, the oldest age class was selected. Additionally, some specimens from zoos exhibited peculiar dental wear due to deviations from the diet

in the wild and therefore, dental wear was not considered significantly valid in such cases. For zoo animals, absolute age data were used whenever these were available. If no data on the age of the animal was available and the relevant sutures were closed, age was mentioned as E+. The same attribution was followed for wild individuals that did not retain an M1.

## Results

Principal Component Analysis (PCA) based on shape data summarized significant skull differences between sexes (male and female) (Fig 3A). Additionally, some distinction between the geographically separated north (*G. camelopardalis*, *G. reticulata*) and south (*G. giraffa*, *G. tippelskirchi*) clades was observed (Fig 3B). The four recognized *Giraffa* species form a gradient that reflects their geographical range (Fig 3C and 3D). The main differences along the axes include the ossicones (frontal and median), orbits, rostrum, nuchal crest, and palate (Fig 3E).

Regression analysis shows that a large portion of skull variation is allometric (31.7% predicted; p < 0.0001) with all the studied specimens following the same trajectory. This shows that the ontogenetic development of giraffe skulls follows a pattern that is strongly affected by allometry. In this sense, ontogeny and allometry are interconnected, influenced by the development of the ossicones. Sexes differ significantly in skull shape and size (p < 0.0001; t-test; Fig 4A and 4E), because males grow larger and, hence, tend to display larger ossicones (frontal and median), broader orbits, and higher nuchal crest, as described in previous studies [7, 8, 17, 18, 50]. Therefore, the centroid size of the adult male skulls is distinctly larger than that of the female ones. Juvenile skulls (represented on the left side of the regression plots) are somewhat more dispersed in the plot than those of the adults, which has been interpreted as evidence for larger cranial parts changing more during the juvenile age stages in other artiodactyls [51, 52]. The ontogenetic development of the ossicones in the first age classes is drastic. Juveniles that are just days/weeks old do not show any signs of ossicones, whereas juveniles 1 or 2 years old have developed ossicones that are not very different in size from those of adult females. This is reflected in the wider dispersion of the juveniles in the CS axis, showing the ontogenetic leaps that take place in the early stages of life. Differences in shape and size between the north and south clades are apparent across the dataset (Fig 4B), as well as in each sex separately (Fig 4C and 4D). The regression scores of the north clade are higher than those of the south clade. This shows that the two forms follow different ontogenetic trajectories that lead to different morphotypes in the adult forms. A more detailed description of these morphological differences is discussed below.

Interestingly, our analysis indicates subtle allometric differences and skull variation (PCA) between the north and south clades, entailing scaling differences between the species (Fig 3). After removing the contribution of allometry to shape data variation (i.e., summarizing the variation of regression residuals, which are devoid of allometry), previously detected cranial differences between the sexes (Fig 5A), the north and south clades (Fig 5B), and the four *Giraffa* species (Fig 5C and 5D) are most evident, whereas the affected cranial regions (Fig 5E) are similar to that of the PCA of the superimposed coordinates. Despite the relative position of the four species aligning with their phylogeny based on the phylogenetic hypothesis of Coimbra et al. (2021) [34], there is some level of overlap (Fig 5D).

To assess and visualize the possible shape differences between the four species, a Canonical Variate Analysis (CVA) was conducted based on the taxonomy of the four species (Fig 6). The analysis found a significant distinction between the respective species (CV1 = 68.9%, and CV2 = 22.6% explained variance, respectively) (Fig 6A and 6D), supported by permutation test p-values of the Procrustes Distances (p < 0.0001 for all pairwise comparisons; S5 Table). The conspecific subspecies represented are relatively homogenous (Fig 6B), while the ossicones and the rostrum length show the main differentiating cranial features (Fig 6C).

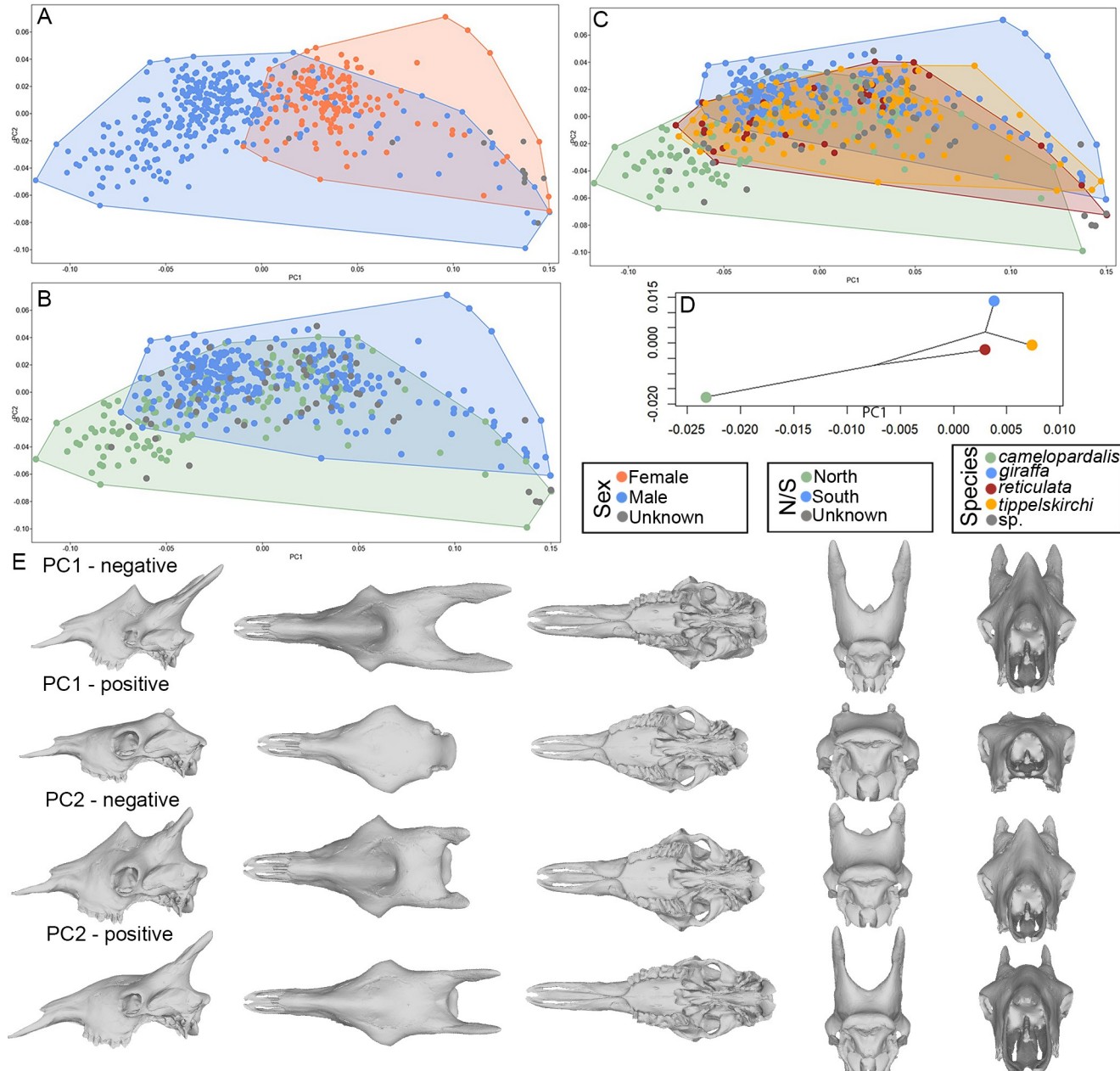

**Fig 3. Principal component analysis on the superimposed coordinates.** (A) PCA ordination using the superimposed coordinates of the complete dataset, based on sex. (B) PCA ordination using the superimposed coordinates of the complete dataset, but colors represent north and south clades. (C) PCA ordination using the superimposed coordinates of the complete dataset, but colors represent species. (D) Phylomorphospace based on the PCA in C. (E) 3D visualization of the shape changes along the two axes of the PCA.

CVA on the subspecies groupings (CV1 = 46.2%, CV2 = 20.2%) shows a more complex pattern, whereby several subspecies of the same species can be distinguished (Fig 6E and 6H), and species distributions are more expanded (Fig 6E and 6I). Importantly, differences between most of the subspecies were statistically significant (p < 0.05; in most cases p < 0.0001), with the exceptions of the *G. c. antiquorum* / *G. c. peralta*, *G. c. antiquorum* / *G. reticulata*, *G. reticulata* / *G. c. peralta*, and *G. g. angolensis* / *G. g. giraffa* pairs (S5 Table). Shape differences observed related to the ossicones, the palate, and the rostrum (Fig 6G).

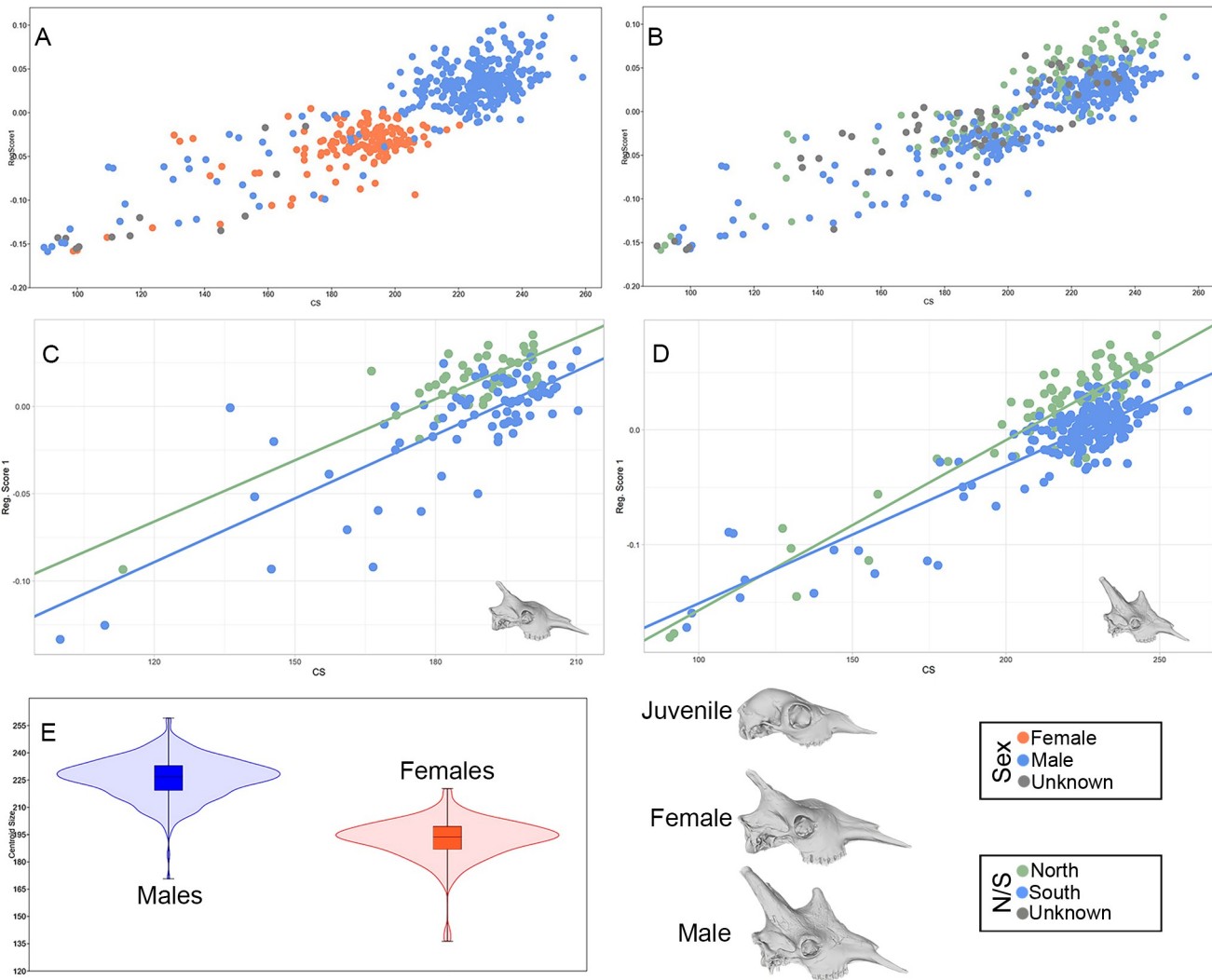

**Fig 4. Allometry, ontogeny and sexual dimorphism in the giraffe skull.** (A) Regression scores of skull shape data (symmetric component, complete dataset) and head size (Centroid Size—CS), showing the different position of males and females along the same trajectory. (B) regression scores of skull shape data (symmetric component, complete dataset) and head size (Centroid Size—CS), but the colors representing north and south clades. (C) regression scores between skull shape data (symmetric component) and CS of females with known taxonomy, showing the difference between north and south clades. (D) regression scores between skull shape data (symmetric component) and CS of males with known taxonomy, showing the difference between north and south clades. (E) violin boxplot showing the head size differences (CS) between adult males and females.

Discriminant Function Analysis (DFA) was used to check pairwise for false positive results of the CVA. The cross-validation substantiated statistically significant discrimination between all species and most subspecies (S5 Table), verifying distinct cranial shape morphologies for most of the studied taxa.

## Discussion

Our findings show a pronounced sexual dimorphism in skull allometry (shape and size) with relatively similar ontogenetic trajectories for the two sexes (i.e., males differ because they grow larger), with the distinction of north and south clades noted in both sexes. The ontogenetic development of the giraffe skull is significantly influenced by allometry. The presented preliminary results suggest that the exact role of allometry in giraffe ontogeny and evolution should

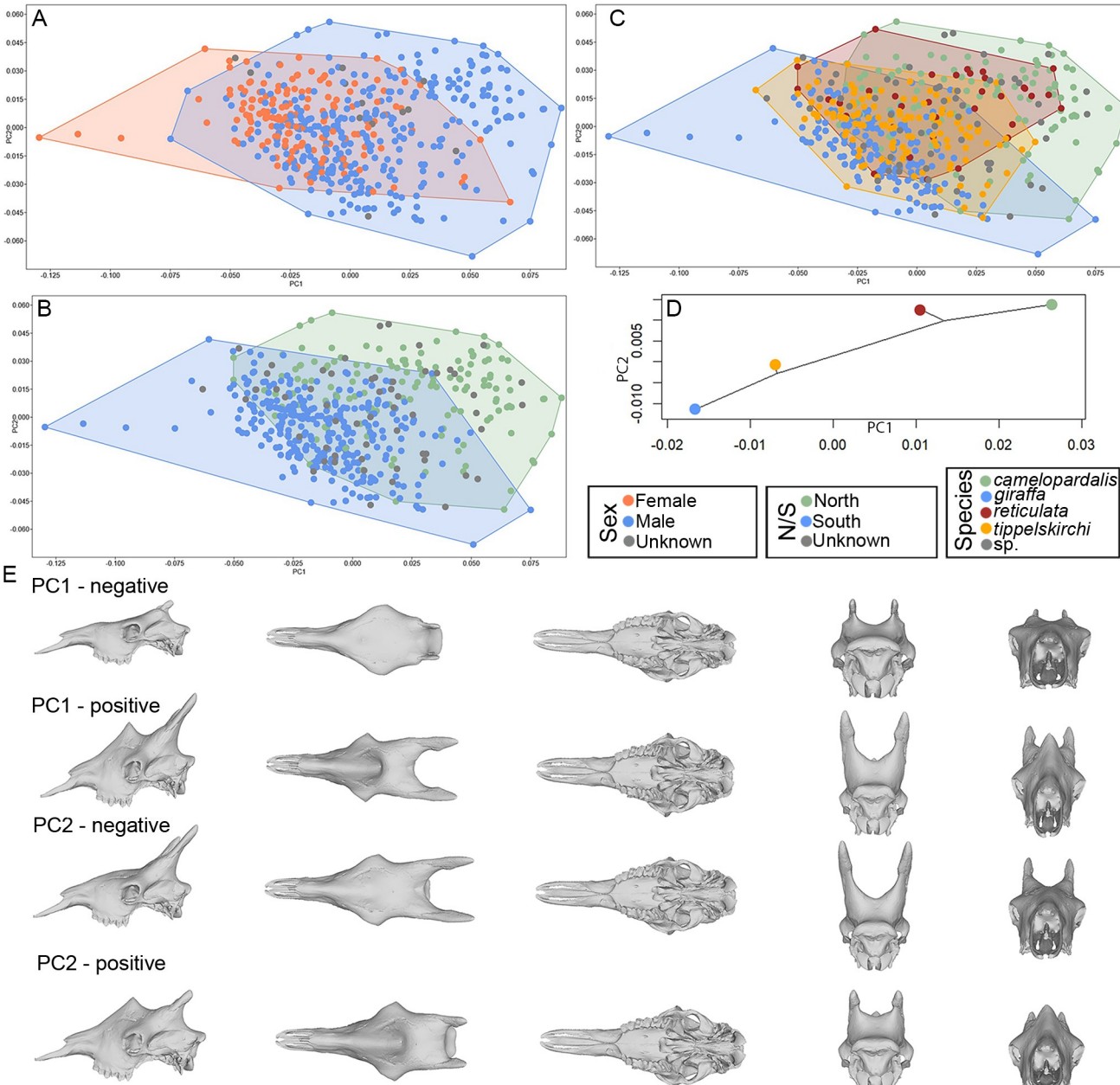

**Fig 5. Principal component analysis on the regression residuals.** (A) Allometry-free PCA using the regression residuals of the complete skull dataset, colored by sexes. (B) Allometry-free PCA using the regression residuals of the complete skull dataset, but colors represent north and south clades. (C) Allometry-free PCA using the regression residuals of the complete skull dataset, but colors represent species. (D) Phylomorphospace based on the PCA in C. (E) 3D visualization of skull shape change along the two axes of the PCA.

be further evaluated. In addition to the ontogenetic differences, there are statistically significant variations in the shape of the distinct *Giraffa* species and most of the subspecies, based on multivariate analyses. The differences mainly concern the ossicones and in particular the development of the median ossicone, which is much more developed in the north clade (Fig 7). In the northern giraffe (Fig 7A), the median ossicone is very high and its dorsal surface is relatively small, resulting in a pointy outline. In the reticulated giraffe (Fig 7B), the median ossicone is relatively high (but lower than in the northern giraffe) and its dorsal surface is

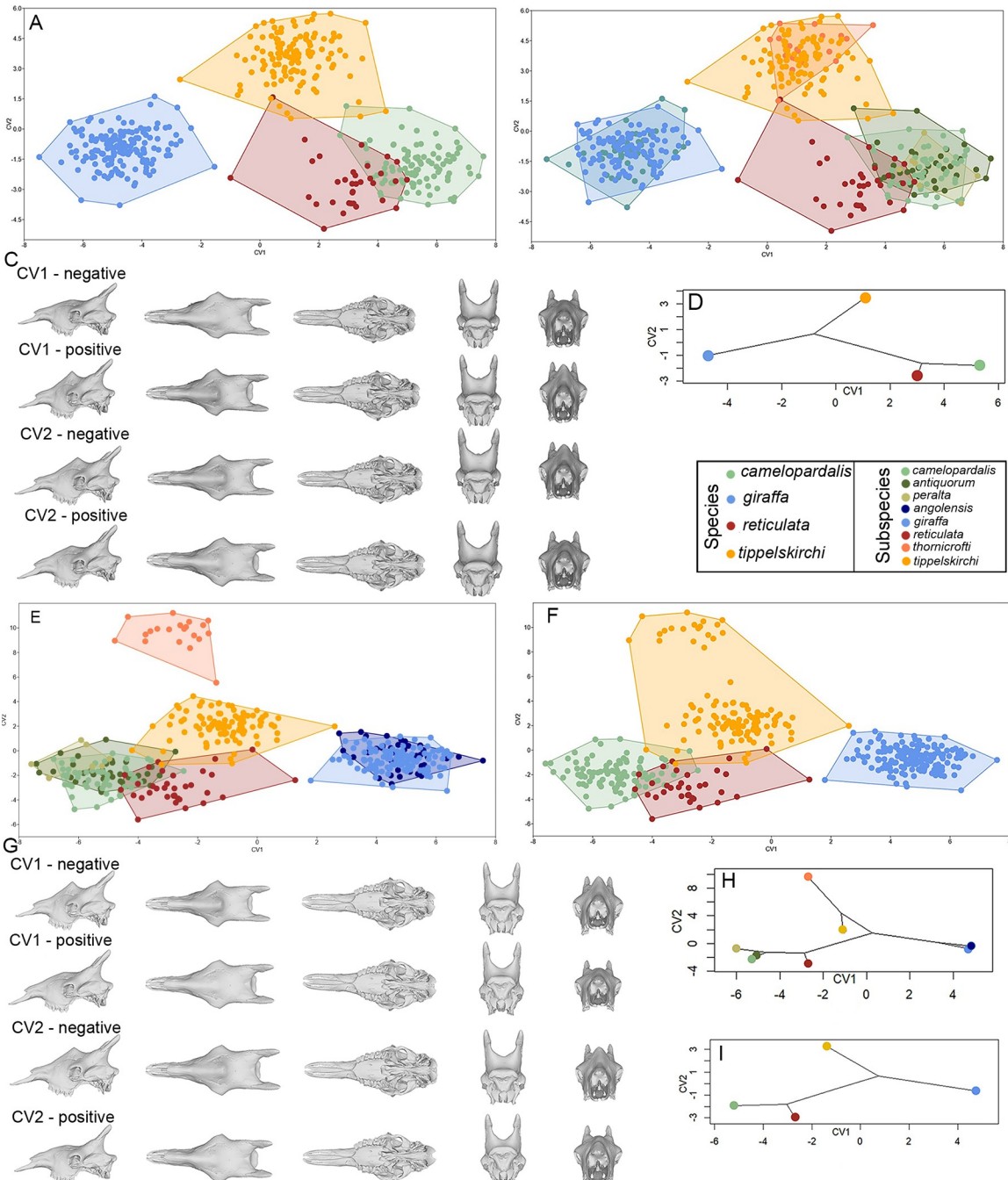

**Fig 6. Canonical variate analysis based on species and subspecies.** (A) Canonical Variate Analysis of the four species. (B) Canonical Variate Analysis of the four species of the four species, but colors represent subspecies. (C) 3D visualization of skull shape change along the two axes of the CVA for the species. (D) Phylomorphospace of the four species based on the plot in A. (E) Canonical Variate Analysis of the eight subspecies. (F) Canonical Variate Analysis of the eight subspecies, but colors represent species. (G) 3D visualization of the shape changes along the two axes of the CVA for the subspecies. (H) Phylomorphospace of the eight subspecies in the CVA shown in E. (I) Phylomorphospace of the four species in the CVA shown in F.

larger, creating a more hill- or step-like outline. In the Masai giraffe (Fig 7C), the median ossi-cone is much smaller than in the two former species, but it is still present, forming a small hill. Finally, the median ossicone of the southern giraffe (Fig 7D) is faint to practically absent, since

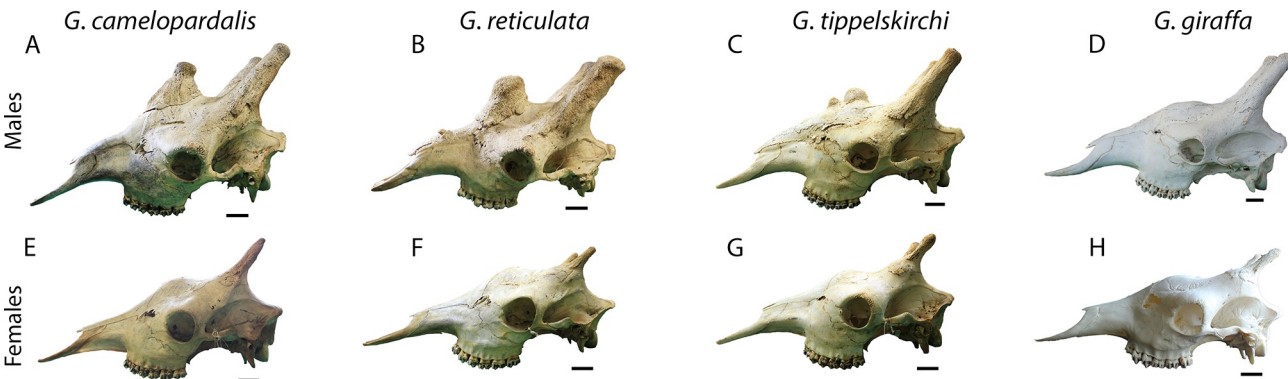

**Fig 7. Comparison of male and female skulls of the four species in lateral view.** (A) USNM-163312, *Giraffa camelopardalis camelopardalis*, male, age class E, Kenya. (B) USNM-182192, *Giraffa reticulata reticulata*, male, age class F, Kenya. (C) FMNH-127881, *Giraffa tippelskirchi tippelskirchi*, male, age class E, Tanzania. (D) ZIM-TCI-319, *Giraffa giraffa giraffa*, male, age class E, Zimbabwe. (E) RMCA-RG.3710, *Giraffa camelopardalis antiquorum*, female, age class F, D. R. Congo. (F) FMNH-32901, *Giraffa reticulata*, female, age class F, Ethiopia. (G) FMNH-127880, *Giraffa tippelskirchi tippelskirchi*, female, age class F, Tanzania. (H) NAM-ING-2164, *Giraffa giraffa angolensis*, female, age class D, Namibia. Scale bars equal 5 cm.

the cranial elevation above the eyes is mostly formed by the inflation of the cranial surface and not by the accumulation of secondary ossification. The differences are milder in females (Fig 7E–7H) with a slight elevation of the median ossicone being observed in the north forms. Each species is represented by a wide morphological range and considerably deviations from this pattern can be seen. However, based on our results, these are some characteristic examples for the standard morphology of the four species. The characteristics of the south clade resemble more the plesiomorphic morphology of fossil *Giraffa* which is related to the role of ossicones in their social behavior [13–17, 38–39].

The IUCN currently recognizes just one species of giraffe, divided in nine subspecies, even though multiple studies over the past 50 years have highlighted the distinctness between different types of giraffe in terms of pelage pattern, cranial and postcranial morphology, genetics, ecology, and behavior [12, 18, 23, 24, 27–37, 53–62]. However, some of these findings were based on limited and not necessarily primary data sets, and not every study incorporated new data. For instance, Hausen [63] analyzed pelage pattern variability in wild giraffe across nine putative subspecies and found it unsuitable for taxonomic distinction due to high variability within and between populations, confirming the results of previous studies [18, 27, 50].

Whether the differences observed are sufficient to justify classifying them as distinct species thus remains a key taxonomic question for modern *Giraffa*. Historically, most studies based their taxonomic conclusions on the biological species concept, which lumps all giraffe to a single species as they hybridize in captivity [27, 28]. However, this concept has limitations, especially concerning fossil species or introgressive hybridization [64]. As more morphological, genetic and ecological data are assembled, the information available is proving to be more influential in our understanding of the concept of species, and currently 16–32 different species concepts exist [65]. While new information takes time to become mainstream, there are several recent examples where the taxonomy of mammals was reviewed and changed, e.g., clouded leopard (*Neofelis*) [66], takin (*Budorcas*) [67], African elephant (*Loxodonta*) [68], reindeer (*Rangifer*) [69], and orangutan (*Pongo*) [70].

Multiple lines of evidence support four distinct extant species of *Giraffa*. Based on the genetic species concept, similar to both the genealogical and phylogenetic species concepts, there are indeed well-structured clusters in molecular phylogenies that show a relatively old age of divergence [18, 26, 28, 29, 31, 34]. Based on the morphological species concept, similar to the phenetic concept, there are also differences between these taxa [10, 18, 21, 24, 28, 54].

This is also true for the reproductive competition species concept [71], which does not focus on the result of the reproduction (a possible hybrid produced by two unrelated taxa) but on the preceding processes of mating. Whilst some *Giraffa* species successfully hybridize in captivity, evidence of hybridization in the wild was predominantly anecdotal and a recent genomic assessment showed that amongst the three *Giraffa* species in Kenya, no recent hybridization has occurred [31, 32, 34, 36]. We propose that it is essential to combine data from multiple approaches to ascertain informed taxonomy. An integrative approach is necessary for modern species evaluations [72], and the present study offers an important additional piece of information in the discussion on giraffe taxonomy.

The need to develop a universally accepted taxonomic scheme for extant *Giraffa* is key to their long-term conservation, and the use of our current cranial morphometric findings coupled with recent genomic and pelage data provides convincing and cohesive data towards this direction. We conclude that the distinction among the four modern *Giraffa* species is verified from cranial shape similar for many subspecies of the same species. Concerning conspecific subspecies, the results show clear definition for *G. t. thornicrofti* and *G. t. tippelskirchi*, as well as between *G. c. camelopardalis* and the other two northern subspecies (*G. c. antiquorum* and *G. c. peralta*), whereas there was no difference between *G. c. antiquorum* and *G. c. peralta* or between *G. g. angolensis* and *G. g. giraffa* (Fig 6 and S5 Table).

Even though taxonomy in its classic sense is a field of biology that was more prominent in the early days of science, it affects many other branches of biology that are crucial for our understanding of the natural environment, most notably conservation. As noted in previous works [1–5, 18], conservation is impacted by taxonomic status. For example, the recognition of the African forest elephant (*L. cyclotis*) as a distinct species from the African savannah elephant (*L. africana*) by the IUCN has been crucial for mobilizing increased and targeted financial and technical support for their independent conservation [65, 73–76]. The formal recognition of four distinct *Giraffa* species is similarly important to avoid admixture and translocations between non-related populations (in the sense of taxonomically distinct groups; not just beyond the range of inbreeding risk) alongside an increased understanding of their ecology and threats. An assessment of the four *Giraffa* species would likely result in increased conservation efforts and actions as three of the four species would probably be listed as endangered or critically endangered on the IUCN Red List of Threatened Species, the world's barometer of life. Ultimately, without a more streamlined system for defining species, many conservation efforts may unknowingly be doing too little, too late.

The present study is based on state-of-the-art methodologies and an unprecedented huge dataset of giraffe skulls. We propose that future developments aim at assessing specific objectives that can further increase our knowledge of giraffe cranial shape. The ontogenetic development and the evolutionary significance of ossicones are key aspects in the understanding of giraffe diversity and the dynamics between the different taxa. The addition of similar approaches for the endocranium, the mandible or postcranial elements may also show taxonomically significant differences that were previously unnoticed due to methodological or sampling limitations. We anticipate that our study will be the base for future research that may uncover other diversity patterns in the skeletal anatomy of the giraffe, enabling us to better understand the physical nature of these animals.

## Conclusions

In summary, the results of the present study suggest that there is in fact considerable difference in cranial shape between the four species of giraffe. Even though the present study is focusing on cranial morphology only, the herein derived classifications contribute to a growing body of

similar evidence reported from many studies [29, 34, 37, 55]. However, regardless of the official taxonomic status of the giraffe lineages, conservation efforts should be focused on distinct forms, avoiding admixtures and translocations between non-related populations. Based on the current results and the consideration of the ossicones as a key trait of the giraffe anatomy and behavior, it is suggested that conservation effort focus on the four distinct species of giraffe, and in some cases even on their subspecies.

## Supporting information

**S1 Appendix. Institutional abbreviations.**
(DOCX)

**S1 Table. Summary of the material studied herein.** M: male, F: female, and X: unknown sex.
(XLSX)

**S2 Table. Complete catalogue of the studied *Giraffa* skull specimens.**
(XLSX)

**S3 Table. Definition of the landmarks used in the current protocol.**
(XLSX)

**S4 Table. Characteristics and actual age of all the age classes followed herein.**
(XLSX)

**S5 Table. Statistical significance between *Giraffa* species and subspecies based on CVA and DFA.** The resulting p-values of the permutation tests for the Procrustes Distances of the CVA and DFA between different species and subspecies. Abbreviations: *ant.* = *antiquorum*, *cam.* = *camelopardalis*, *per.* = *peralta*, *ang.* = *angolensis*, *gir.* = *giraffa*, *ret.* = *reticulata*, *tho.* = *thornicrofti*, *tip.* = *tippelskirchi*. Statistically significant p-values ($< 0.05$) were written in bold.
(XLSX)

## Acknowledgments

We are grateful to many people and institutions for their help in this study that collectively provided access to collections, additional insight and discussions on the project (in alphabetical order of the institutes): L. Caspers (AMNH), E. Hoeger (AMNH), M. Surovy (AMNH), S. Roussiakis (AMPG), C. Fraticelli–African Parks Network (CHA-ZAK), A. Antonites (DNMNH), H. Fourie (DNMNH), A. Ferguson (FMNH), L. Johnson (FMNH), A. Feijó (FMNH), V. Simeonovski (FMNH), G. Csorba (HNHM), Etosha Heights Private Reserve (NAM-ETO), S. Bean (NAM-ING), M. Egerer (NAM-NYA), Tau Taxidermy (NAM-TAU), P. Kamminga (NCB), P. Kokkini (NHMUK), M. J. Fitzpatrick (NHMZ), K. Gagisa Mkhwananzi (NHMZ), S. Broadley (NHMZ), K. N. Langwane (NHMZ), T. Musareyana (NHMZ), F. Zachos (NMW), A. Bibl (NMW), M. Rotonda (RMCA), E. Gilissen (RMCA), J. Opperman (SAM), P. Prins (SA-SI), I. Ruf (SMF), K. Krohmann (SMF), S. Merker (SMNS), C. Leidenroth (SMNS), M. Sečanský (SNM), Dr V. Jansky (SNM), Uganda Wildlife Authority (UGA-MUR), D. Lunde (USNM), T. Hsu (USNM), F. Otten–Zambia Carnivore Program (ZAM-LUA), E. Bärman (ZFMK), A. Pugh (ZIM-TCI), E. Valakos (ZMUA), F. Mayer (ZMB), K. Mahlow (ZMB), M. Gärtner (ZMB), and A. van Heteren (ZSM). We would like to thank L. Pandolfi and L. Costeur for providing the 3D scans of the specimens housed in NMB, as well as P. Prins for providing the scans of the specimens from SA-SI. Lastly, we would like to thank the Giraffe Conservation Foundation for their financial and technical support in making this study possible, and the Department of Biological Sciences at the University of Cape Town for hosting the

project. We are thankful to the editorial board of PLoS ONE and to the three reviewers (N. Solounias and two anonymous reviewers) for their help during the peer review and the production process.

## Author Contributions

**Conceptualization:** Nikolaos Kargopoulos, Jesús Marugán-Lobón, Anusuya Chinsamy, Stephanie Fennessy, Fredrick Lala, Moses Otiende, Julian Fennessy.

**Data curation:** Nikolaos Kargopoulos.

**Formal analysis:** Nikolaos Kargopoulos, Jesús Marugán-Lobón.

**Funding acquisition:** Stephanie Fennessy, Julian Fennessy.

**Investigation:** Nikolaos Kargopoulos, Jesús Marugán-Lobón, Anusuya Chinsamy.

**Methodology:** Nikolaos Kargopoulos, Jesús Marugán-Lobón.

**Resources:** Bernard R. Agwanda, Michael Butler Brown, Stephanie Fennessy, Sara Ferguson, Rigardt Hoffman, Fredrick Lala, Arthur Muneza, Ogeto Mwebi, Moses Otiende, Alice Petzold, Sven Winter, Abdoul Razack Moussa Zabeirou, Julian Fennessy.

**Supervision:** Jesús Marugán-Lobón, Anusuya Chinsamy, Stephanie Fennessy, Julian Fennessy.

**Visualization:** Nikolaos Kargopoulos.

**Writing – original draft:** Nikolaos Kargopoulos, Jesús Marugán-Lobón, Anusuya Chinsamy, Stephanie Fennessy, Julian Fennessy.

**Writing – review & editing:** Nikolaos Kargopoulos, Jesús Marugán-Lobón, Anusuya Chinsamy, Stephanie Fennessy, Julian Fennessy.

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
