## [Decision Letter · Decision Letter 0]

17 Oct 2024

PONE-D-24-31082Heads up – Four Giraffa species have distinct cranial morphologyPLOS ONE

Dear Dr. Kargopoulos,

Thank you for submitting your manuscript to PLOS ONE. After careful consideration, we feel that it has merit but does not fully meet PLOS ONE’s publication criteria as it currently stands. Therefore, we invite you to submit a revised version of the manuscript that addresses the points raised during the review process.

 The three reviewers have provided a number of suggestions that could substantially improve the quality of the manuscript. In particular, I agree with reviewer 2 that some aspects of the results and methods could be clarified (particularly relating to sample size). Finally, 2 of the reviewers noted that you have not made the data and code underlying your results publicly available. This is an obligatory condition of publishing in PLOS ONE, so please make sure to provide a link to your publicly deposited data and code repository in your revised manuscript.

We look forward to receiving your revised manuscript.

Kind regards,

Joel Harrison Gayford

Academic Editor

PLOS ONE

Journal Requirements:

2. In your manuscript, please provide additional information regarding the specimens used in your study. Ensure that you have reported human remain specimen numbers and complete repository information, including museum name and geographic location. 

For more information on PLOS ONE's requirements for paleontology and archeology research, see https://journals.plos.org/plosone/s/submission-guidelines#loc-paleontology-and-archaeology-research.

Reviewers' comments:

Reviewer's Responses to Questions

**Comments to the Author**

1. Is the manuscript technically sound, and do the data support the conclusions?

Reviewer #1: Yes

Reviewer #2: Partly

Reviewer #3: Yes

2. Has the statistical analysis been performed appropriately and rigorously? 

Reviewer #1: I Don't Know

Reviewer #2: Yes

Reviewer #3: I Don't Know

3. Have the authors made all data underlying the findings in their manuscript fully available?

Reviewer #1: Yes

Reviewer #2: No

Reviewer #3: No

4. Is the manuscript presented in an intelligible fashion and written in standard English?

Reviewer #1: Yes

Reviewer #2: Yes

Reviewer #3: Yes

5. Review Comments to the Author

Reviewer #1: I cannot evaluate their technical methods and statistics. The study is unique and number of skulls is huge and worth publishing. They know there are hybrids hiding in the data but overall the patterns would remain because the number of skulls is large.

also44 Features of display – explain what this means

56 - 57 the ossicones of Palaeomeryx (Sansan and Aternay) – Libytherium Sivatherium Giraffokeryx Honanotherium resemble those of Giraffa. – the okapi ossicone is different from Giraffa.

74 replace intrasexual with sexual dimorphism

93 the entire study depends on this assumption

:

Victoria Lee, Nikos Solounias 2023. Osteological foot differences between giraffes from distinct geographical locations; a pilot study. Zoologishcer Anzeiger https://doi.org/10.1016/j.jcz.2022.01.001 - We got the cover too.

The giraffe: parts unknown for Cambridge Scholars Press - 2023. Book With 60 plates. Nikos Solounias

Putting Samotherium in its place. Cambridge Scholars Press. Book

Reviewer #2: The authors present a research article focused on the applications of geometric morphometrics to quantify shape variation in Giraffe craniums. They use this as a basis to analyse differences between (sub)species, sex and relate this to ontogenetic trajectories. The findings of the paper show that there are differences between the north and south clades, with statistics showing significant differences in shape between the first four species. They relate this to differences in display features and suggest that the findings may be useful for taxonomy and conservation.

Firstly, I like the title as it adequately provides insight into the overall result, though I think this could be related to the importance of the findings (i.e. to conservation). This leads into the abstract in lines 33-35, where I think you could provide a more robust reasoning as to why it is important to understand their evolution, ontogeny and taxonomy. Later in the abstract you tend to focus on conservation and mention taxonomy in the opening lines of the main text, so I would use this as the motivation to then set up the rest of the abstract. The methods, results and importance in the abstract are concise and easy to follow (lines 35-45).

In the introduction, I do not think that line 55-56 is needed, I think it should be clear that the study is important without saying it explicitly. Instead use this sentence to say why your study offers for an increased understanding of distinguishing between taxa. You provide some good reason for importance in the following sentences so maybe use these (lines 58-66). You could also use this as the motivation in the abstract. Good reasonings for ecological importance are also denoted in lines 68-70. Lines 73-76 is a little confusing, you say our results, does this refer to this study or another? If a previous one, please cite the study and make this clearer, if this one then I do not think it should be discussed here. Instead focus on setting up what previous studies have done, and then discuss their weaknesses and use this as the motivation as to why a more thorough 3D shape analysis is needed. You give this motivation lines 71-73 and 76-79, set up the differences in findings to then say we used a 3DGM approach to overcome these issues.

In the methods, I do not think you need table 1 in the main text, I would move this to the supplementary, the shape figures (Fig.1) show the distributions in a round about way. I think that it is great that you included this, but it fulfils a similar role to table S1, and it is not integral to understanding the paper. The abbreviations in the text in lines 98-124 makes it difficult to navigate the text. In lines 88-89, you mention they were scanned, but then provide more details in the next section (methods). I would integrate materials and methods. A way to circumvent the issue is do not mention scans in line 88, instead say collected. I have no issues with descriptions of the way of the scanning was performed.

You provide details of the landmark scheme in Table S2 which is good, but I suggest that you provide a figure that shows the landmark scheme. I would replace Table 1 with this figure. You could for instance provide a landmark scheme on a single specimen, where you show the landmarks and semilandmarks on the underlying mesh. You mention you use two segments (lines 141-142), so I would give a different colour scheme for each of these modules. Also given this, have you thought about looking at the phenotypic integration and modularity in the skulls of these two regions? The description in how the landmarks were used are good, with the software and packages and their versions clearly demonstrated, but I would like more details about the CVA and DFA particularly to make the study more repeatable.

The next part of the methods is related to ontogeny. I like the idea, but I would question if you had enough repeated specimens across all of the different taxa to make a meaningful insight. You give the age classes in Table S3, but you do not provide the actual number of specimens for each (sub)species that you use, so this makes inference about the meaningfulness of these results difficult. I have no issue with the use of the dental wear, rather my concerns are about sample size.

For the results, I agree with the findings that there are differences between sexes and the clades, but I think you need to be careful with one and the other affecting the other. Instead of having two separate plots, instead I recommend making a single plot that includes data related to sex and geography, this way you can remove the bias of sex on the results for geography (if more males in one than the other) and vice versa. Figure 1 is very busy, so I have some suggestions. Condense to two subplots: (1) a PCA with classes showing both species and sex, (2) a PCA with geography and sex. Then I would just display the extreme morphologies using one view, rather than 5. This will help keep it easier to follow. Providing the landmarks on the craniums is also useful here to back up statements in lines 187. Could heatmaps using thin plate splines be used to further demonstrate these statements?

In lines 195-203, you then discuss allometry and relate this to ontogeny. You mention juveniles, but I cannot identify them in the plot. I think it may be best to just focus on the allometry side of things using adults only to understand the results better. I think the statements in lines 202-203 related to geography are hard to see, I think this needs some rethinking of how to show this. The differences between sexes are clearer. Could you do plots instead related to (1) sex and (2) species?

I think figure 3 does help to back up your points in 215-216, but I would argue that allometry is an important part of the biology and removing its affect does not make as much sense. Using landmarks removes the size variation, whilst still allowing to compare allometric differences, I therefore recommend not focusing on the results in figure 3 as much as currently. Instead let the CVA which you have performed to demonstrate that the species and subspecies can easily be identified using this method, which I think is clear giving figure 4. I like figure 4 for this, but I still have some of the same issues I had with figure 1 with it being too busy. Could perhaps give an example of each species (mesh or landmarks) next to the convex hull for each species. Table 1 reaffirms the results of these, but it is quite messy and large so may be better suited for the supplementary as Figure 4 shows a lot of these differences. Maybe keeping it just for species (rather than subspecies too) may a way of keeping Table 1 in the main text without it being too hard to read.

In lines 266-267 you highlight the differences between sexes being related to allometry which I think re-iterates the points on figure 3. I agree with the findings, but I would maybe urge to focus more on the allometry than the ontogeny (given previous remarks on sample size). Differences in the shape between species are clear in the morphospace plots, and you mention this is mainly due to the ossicones, I think you need a figure somewhere that really demonstrates this (visualising the landmarks will help). Good reasoning behind the differences is given in lines 272-273, but then go on to discuss other topics. I would focus on these differences in the following lines (lines 274 to 279). I would then when discussing the taxonomic issue state what others have done (akin to lines 274 to 279), and then mention how ossicones are also useful. This then can reaffirm your statements in lines 301-305 which are great. Think about a best way to argue your points about why your study is important here, rather than contextualising outside of the study (always relate back to your findings).

Lines 306-308 show the importance of understanding the taxonomy of giraffes, and the paper is set up for this. Lines 309-312 do a good job of demonstrating why cranial shape is important for these taxonomic distinctions. Points in lines 320-329 also back up these points.

The conclusions importantly highlight that there are differences in cranial shapes between the giraffes (and previously relate to ossicones). The points in the discussion are relevant given a focus on taxonomy.

Overall, I like the idea of the study and the dataset is of large size given the use of high-res landmarking. I, therefore, think the study has lots of potential to be impactful and has some clear outcomes with differences in morphologies between species, sex and geography and this is very neat. I think the paper can be improved easily by improving the visualisations, something that shows the landmarks and highlights the areas of most variation in the cranium are good starts. Improvements to the PC plots will also make differences clearer and will help clarify the roles of sex and species in determining overall morphology.

It would also be great if you uploaded your code and data to a GitHub repo to help reproduce the results in the future. The landmark data for all specimens also needs to made available for future use (either GPA transformed or base).

A couple of additional points outside of the manuscript, would be to think about what you can show given the obtained data. You have lots of shape data and a phylogeny, I think some simple evolutionary studies could be done to give the paper a wider reach and interest. For instance, you could look at phylogenetic signal using geomorph, you could also look at morphological disparity and evolutionary rates across the different species to further aid in understanding the evolution (and by extension the taxonomy) of these organisms. The inclusion of two modules also means integration and modularity can be further added to give more insights into the roles of ossicones in determining differences. Just a few things to think about if you wanted to improve the study with what I think would be little extra work given what is presented here.

Reviewer #3: The work carried out is highly relevant, with significant implications for the taxonomy, ecology, and conservation of the group. The sampling is particularly impressive, considering that large animals are often difficult to store in natural history museums. The writing is clear and concise, though I believe some sections could be expanded to better discuss the broader impacts of these findings. My suggestions are minimal and mainly focus on elaborating on a few explanatory aspects and enhancing the discussion.

1. Page 4, line 96: Considering that locality was used as a criterion for taxonomic identification and is an important factor in the debate the article aims to address, I believe that a map showing the distribution of the species and subspecies would greatly enhance the study's clarity.

2. Page 5, line 128: Even though the CT scan images are not the primary focus of the study, I think it would be good practice to share the CT scan parameters.

3. Page 6, line 137: I believe an illustrative diagram showing the landmarks would be a great complement to Table 2 in the supporting information. Since you used Stratovan Checkpoint, this might be a bit more challenging to illustrate, but I think it's achievable and would add clarity.

4. Page 6, line 159: You should specify which reference phylogeny you used to create the trees in Mesquite.

5. Page 8, line 191: Instead of writing "same as" I suggest writing out the full legend again for clarity, even if it makes the reading a bit repetitive. This apply to the other legends as well.

6. Page 9, line 222: Again, you need to indicate which phylogeny you're referring to in this section.

7. Page 12, line 272: The sentence, "The characteristics of the south clade resemble more the plesiomorphic morphology of fossil Giraffa, which is related to the role of ossicones in their social behavior," needs more context. References to a paper on the morphology of fossil giraffes and a better explanation of the role of ossicones in their social behavior would add clarity.

8. In the discussion, I believe you could more clearly explain how the IUCN treats the taxonomy of giraffes. It is implied that the IUCN recognizes only one species, but the recognition of subspecies is less clear and could benefit from further elaboration.

9. I also consider it important to dedicate a section before the conclusion to address the limitations of the study (e.g., the issue of giraffe age mentioned in the methodology) and how future studies could overcome these limitations. Since you will be making the CT scan images available, it would also be worth pointing out that other aspects, such as ecomorphology, could be explored in future research.

10. In your acknowledgments, you cite the Giraffe Conservation Foundation for their financial support. Should this not be placed in the financial disclosure section?

6. PLOS authors have the option to publish the peer review history of their article (what does this mean?). If published, this will include your full peer review and any attached files.

Reviewer #1: **Yes: **Nikos Solounias

Reviewer #2: No

Reviewer #3: No

---

## [Author Response · Author response to Decision Letter 0]

31 Oct 2024

We are very thankful to the editorial board of the journal and the reviewers for their fruitful comments. Please find attached a detailed list of the changes that we made.

---

## [Decision Letter · Decision Letter 1]

15 Nov 2024

PONE-D-24-31082R1Heads up – Four Giraffa species have distinct cranial morphologyPLOS ONE

Dear Dr. Kargopoulos,

Thank you for submitting your manuscript to PLOS ONE. After careful consideration, we feel that it has merit but does not fully meet PLOS ONE’s publication criteria as it currently stands. Therefore, we invite you to submit a revised version of the manuscript that addresses the points raised during the review process.

 Thank you for taking the time and effort to revise your manuscript. As you will see, the reviewers are mostly satisfied with the adjustments made, but have a few additional suggestions to improve the manuscript before it can be accepted.  Please submit your revised manuscript by Dec 30 2024 11:59PM. If you will need more time than this to complete your revisions, please reply to this message or contact the journal office at plosone@plos.org. Please include the following items when submitting your revised manuscript:A rebuttal letter that responds to each point raised by the academic editor and reviewer(s). You should upload this letter as a separate file labeled 'Response to Reviewers'.A marked-up copy of your manuscript that highlights changes made to the original version. You should upload this as a separate file labeled 'Revised Manuscript with Track Changes'.An unmarked version of your revised paper without tracked changes. You should upload this as a separate file labeled 'Manuscript'.If applicable, we recommend that you deposit your laboratory protocols in protocols.io to enhance the reproducibility of your results. Protocols.io assigns your protocol its own identifier (DOI) so that it can be cited independently in the future. For instructions see: https://journals.plos.org/plosone/s/submission-guidelines#loc-laboratory-protocols. Additionally, PLOS ONE offers an option for publishing peer-reviewed Lab Protocol articles, which describe protocols hosted on protocols.io. Read more information on sharing protocols at https://plos.org/protocols?utm_medium=editorial-email&utm_source=authorletters&utm_campaign=protocols.

We look forward to receiving your revised manuscript.

Kind regards,

Joel Harrison Gayford

Academic Editor

PLOS ONE

Journal Requirements:

Reviewers' comments:

Reviewer's Responses to Questions

**Comments to the Author**

1. If the authors have adequately addressed your comments raised in a previous round of review and you feel that this manuscript is now acceptable for publication, you may indicate that here to bypass the “Comments to the Author” section, enter your conflict of interest statement in the “Confidential to Editor” section, and submit your "Accept" recommendation.

Reviewer #2: All comments have been addressed

Reviewer #3: All comments have been addressed

2. Is the manuscript technically sound, and do the data support the conclusions?

Reviewer #2: Yes

Reviewer #3: Yes

3. Has the statistical analysis been performed appropriately and rigorously? 

Reviewer #2: Yes

Reviewer #3: I Don't Know

4. Have the authors made all data underlying the findings in their manuscript fully available?

Reviewer #2: No

Reviewer #3: Yes

5. Is the manuscript presented in an intelligible fashion and written in standard English?

Reviewer #2: Yes

Reviewer #3: Yes

6. Review Comments to the Author

Reviewer #2: The authors have made a significant effort to address my comments from the previous review. Below are a few comments to help further improve the manuscript:

1. The institutional abbreviations may be better suited in the supplementary material rather than main text, unless the authors think it is essential to contain in the main text. They have discussed this comment, so I will delegate to the editor in terms of journal fit.

2. The authors have kept the majority of the shape figures the same. I suggested to reduce the size of these figures in terms of the morphological representations, the authors have justified why these have not been changed. I do not think the figures are hard to navigate by any means, therefore, I believe the style of these is best left to the authors and editors.

3. The authors have made a significant effort to improve their data accessibility with inclusion of more data and code. I suggest adding the data to the GitHub page upon publication, but I think for publication this is vastly improved. I would still like for the shape data (landmark co-ordinates, either Base or GPA transformed) to be made available for reproducing the results and plots before publication.

4. I still have some questions in referring to ontogeny, the authors have clarified in their response of how they use the samples, but I still wonder if you can thoroughly look at ontogeny. Given it is something discussed minimally in the text, I would omit its use and stick with a focus on allometry. I have no problem retaining the juveniles in relation to this.

Overall, I still think some additional data, especially the landmark co-ordinates need to be made available before publication. Stats from further downstream analysis are less important, but I think any tables generated from this also should be made available. I still have some concerns around the use of ontogeny and its place in the paper, which was something also mentioned by the editor. I think the authors should try and address this more thoroughly too. Otherwise, I am happy with all the changes made and believe the authors have performed some great work here.

Reviewer #3: I believe that all the issues I raised in the review have been addressed. The revised text is much clearer, and the new figures added are great. The only new suggestion I would have is on page 6, line 165, and page 9, line 232. I would replace "phylogenetic scheme" with "phylogenetic hypothesis," but this is merely a stylistic and optional choice. I recommend the acceptance of the manuscript.

7. PLOS authors have the option to publish the peer review history of their article (what does this mean?). If published, this will include your full peer review and any attached files.

Reviewer #2: No

Reviewer #3: No

---

## [Author Response · Author response to Decision Letter 1]

19 Nov 2024

Our comments can be found in the Rebuttal letter attached

---

## [Editor Report · Decision Letter 2]

20 Nov 2024

Heads up – Four Giraffa species have distinct cranial morphology

PONE-D-24-31082R2

Dear Dr. Kargopoulos,

We’re pleased to inform you that your manuscript has been judged scientifically suitable for publication and will be formally accepted for publication once it meets all outstanding technical requirements.

Kind regards,

Joel Harrison Gayford

Academic Editor

PLOS ONE

---

## [Editor Report · Acceptance letter]

27 Nov 2024

PONE-D-24-31082R2 

PLOS ONE

Dear Dr. Kargopoulos, 

I'm pleased to inform you that your manuscript has been deemed suitable for publication in PLOS ONE. Congratulations! Your manuscript is now being handed over to our production team.

Kind regards, 

on behalf of

Mr. Joel Harrison Gayford 

Academic Editor

PLOS ONE